# Nanoemulsions for Enhancement of Curcumin Bioavailability and Their Safety Evaluation: Effect of Emulsifier Type

**DOI:** 10.3390/nano11030815

**Published:** 2021-03-23

**Authors:** Raquel F. S. Gonçalves, Joana T. Martins, Luís Abrunhosa, António A. Vicente, Ana C. Pinheiro

**Affiliations:** Centre of Biological Engineering, University of Minho, 4715-057 Braga, Portugal; raquel.goncalves@ceb.uminho.pt (R.F.S.G.); joanamartins@deb.uminho.pt (J.T.M.); d3024@deb.uminho.pt (L.A.); avicente@deb.uminho.pt (A.A.V.)

**Keywords:** nanoemulsions, lecithin, rhamnolipids, Tween^®^ 80, in vitro static digestion, cytotoxicity

## Abstract

This work aimed at evaluating the effects of different emulsifiers on curcumin-loaded nanoemulsions’ behavior during digestion, its safety and absorption, to develop nanoemulsions that provide safety and improved curcumin functionality. Nanoemulsions (NEs) were produced using two bio-based (lecithin (LEC) and rhamnolipids (RHAM)) and one synthetic (Tween^®^80 (TWE)) emulsifier at similar concentrations. Different NEs were subjected to in vitro digestion. The cytotoxicity and permeability tests were performed in Caco-2 cells. NE_TWE were stable during all phases of in vitro digestion, whereas NE_LEC and NE_RHAM were found to be unstable from the gastric phase. NE_TWE showed 100% of free fatty acids released, followed by NE_RHAM and NE_LEC. Curcumin’s bioaccessibility and stability increased in the following order: NE_LEC > NE_RHAM > NE_TWE. NE_LEC and NE_TWE did not show cytotoxic effects in any of the concentrations tested, while NE_RHAM presented high cytotoxicity in all concentrations tested. The apparent permeability coefficients were determined for NE_LEC and NE_TWE; however, the results were not statistically different. These results showed that the emulsifier used has a high impact on nanoemulsions’ behavior under the digestion process and on their cytotoxicity. This work contributed to the state-of-the-art’s progress on the development of safer curcumin delivery systems with improved functionality, particularly regarding the proper selection of ingredients to produce said systems.

## 1. Introduction

The interest in the application of nanotechnology to foods has grown rapidly in the last decade, mainly in the areas of functional foods, food packaging, and food safety [1]. In particular, the development of functional foods has been increasing due to the rise in consumers’ awareness about the impact of diet on their health [2]. Furthermore, the growing interest in the use of sustainable and “label-friendly” ingredients is promoting the replacement of synthetic ingredients by bio-based alternatives [3].

Curcumin is the principal curcuminoid present in turmeric, *Curcuma longa,* and it is commonly used in food as a pigment and spice. This polyphenol has several reported health benefits, such as antioxidant, anti-microbial, anti-inflammatory, and anti-tumoral effects [4]. However, curcumin presents low solubility in aqueous solutions, sensitivity to light, and low bioavailability, which limit its incorporation into food products [4]. In order to overtake these limitations, various delivery systems capable of encapsulating this phenolic compound have been studied, such as nanoemulsions (NEs) [5], liposomes [6], excipient emulsions [7], polysaccharide nanoparticles [8], and nanogels [9].

NEs are one of the most used colloidal delivery systems for lipophilic bioactive compounds’ encapsulation. They are formed by oil droplets dispersed in an aqueous solution and stabilized with an emulsifier, producing particles with sizes between 20 nm and 200 nm [10]. Due to their high physical stability, good dispersibility, easy production, low opacity, and high surface area, NEs are used to improve the oral bioavailability of different bioactive compounds [2,11]. Lipid carrier, emulsifier type, and droplet size are the most influential factors determining the efficiency of micelle formation and its uptake by intestinal cells, thus influencing the bioactive compounds’ bioaccessibility [12]. Regarding the oil carrier, it has been observed that triglyceride chain length has a huge influence on the lipid digestion and on bioactive compound bioaccessibility [12]. Some authors observed that the length of triglyceride chains is positively correlated with the bioaccessibility of certain bioactive compounds. It has been postulated that long-chain fatty acids increase the solubilization capacity of eugenol [12], vitamin D_3_ [13], and β-carotene [14,15].

Emulsifiers are crucial to NE formation, because they promote physical stability and help in their formation [3]. Various natural emulsifiers can be used in the food industry, including proteins, polysaccharides, biosurfactants, and phospholipids. Lecithin (LEC) is a natural emulsifier extracted from egg yolk, milk, soybean, rapeseeds, or sunflower kernels and is composed of different phospholipids, such as phosphatidylcholine, phosphatidylethanolamine, and phosphatidylinositol; see Figure 1 [16]. The mixture of different percentages of these phospholipids alters LEC’s properties as an emulsifier, i.e., its effectiveness in terms of emulsion formation and stability [3]. NEs using LEC as an emulsifier or co-emulsifier have been successfully produced to encapsulate different bioactive compounds, such as eugenol [17], curcumin [18], vitamin E [19], and omega-3 oils [20]. Furthermore, LEC is also used in several types of nanostructures, such as liposomes [21], pickering emulsions [22], or solid lipid nanostructures [23], as an emulsifier or co-emulsifier. Rhamnolipids (RHAMs) are glycolipids obtained through fermentation processes from some microorganisms, such as *Pseudomonas aeruginosa*. The composition and properties of this biosurfactant can be controlled by changing the strain, substrate, and fermentation conditions. They are constituted by one or two rhamnose units and a non-polar fatty acid chain that contains β-hydroxylalkanote; see Figure 1 [24]. Only a few studies have been conducted to evaluate the formation and stability of NEs using RHAMs as an emulsifier or co-emulsifier [25,26]. There are some studies showing that RHAMs are capable of enhancing epithelial permeability [27,28,29]. However, the influence of this biosurfactant in bioactive compounds’ bioaccessibility, cellular permeability, and toxicity, when used in delivery systems, has not been evaluated. Tween^®^ 80 (TWE) is a synthetic, non-ionic surfactant generally recognized as safe (GRAS), which is composed of a non-polar fatty acid group esterified to a polar polyoxyethylene sorbitan group; see Figure 1. Due to its non-ionic nature, TWE promotes stability over a broad pH range and ionic strength values. However, it presents some instability at temperatures close to the phase inversion temperature [30]. TWE is widely applied as a surfactant to stabilize several types of nanostructures, such as NEs [31], nanostructured lipid carriers [32], solid lipid nanoparticles [33], and liposomes [34].

Some studies have been performed in order to evaluate the influence of bio-based and synthetic emulsifiers on NE behavior throughout the gastrointestinal (GI) tract and on the bioaccessibility of different bioactive compounds. Gasa-Falcon et al. (2019) evaluated the β-carotene bioaccessibility in NEs produced using different emulsifiers (Tween^®^ 20, lecithin, sodium caseinate, and sucrose palmitate). These authors observed that the emulsifier type and concentration have an impact on β-carotene bioaccessibility [35]. Ma et al. (2019) assessed the influence of different emulsifiers (whey protein isolate, modified lecithin, and gum arabic) on fucoxanthin bioaccessibility and also reported that the type and concentration of emulsifier has an influence on several digestion parameters, such as lipid digestion and fucoxanthin bioaccessibility [36]. However, such a complete study on the effect of different emulsifiers on curcumin’s NE behavior under in vitro digestion/absorption, particularly on curcumin’s bioaccessibility and permeability, and on curcumin’s NE cytotoxicity has not been performed before.

The main objective of this study is to provide new insights into the development of safe NEs with maximized curcumin bioactivity (by increasing its bioavailability). For the first time, this work shows a complete evaluation of the digestion/absorption behavior and safety of curcumin NEs produced using various emulsifiers (two natural and one synthetic emulsifier). In particular, the influence of emulsifier type (LEC, TWE, and RHAM) on curcumin bioaccessibility and stability during digestion has been assessed using the harmonized static in vitro digestion protocol. The cytotoxicity and cell permeability across a Caco-2 cell monolayer in all NE formulations have been evaluated.

## 2. Materials and Methods

### 2.1. Materials

PHOSPHOLIPON^®^ 80H, composed of hydrogenated phospholipids from soybean, with 70% phosphatidylcholine, was kindly provided by Lipoid (Steinhausen, Switzerland). RHAM with 90% purity was purchased from AGAE Technologies (Corvallis, OR, USA). Corn oil (LCT) was purchased in the local market (Braga, Portugal). Curcumin, pepsin from porcine gastric mucosa (≥2500 U mg^−1^), bile extract porcine, pancreatin from porcine pancreas (8× USP), Pefabloc^®^ SC and the salts used for preparation of oral, gastric, and intestinal electrolyte solutions, Nile red, and dimethyl sulfoxide (DMSO) were purchased from Sigma-Aldrich (St. Louis, MO, USA). TWE was obtained from Panreac (Barcelona, Spain), sodium hydroxide was purchased from JMGS (Odivelas, Portugal), and hydrochloric acid was obtained from CHEM-LAB (Belgium). Acetonitrile and chloroform were obtained from Fisher Scientific (Hampton, NH, USA). Dulbecco’s modified Eagle’s medium (DMEM), non-essential amino acids, phosphate-buffered saline (PBS), and Hanks’ Balanced Salt Solution (HBSS) were obtained from Lonza (Basel, Switzerland). Penicillin/streptomycin (PS), trypsin-EDTA, CelLytic™ Cell Lysis Reagent, and 3-(4,5-dimethylthiazol-2-yl)-2,5-diphenyltetrazolium bromide (MTT) were purchased from Sigma-Aldrich (St. Louis, MO, USA); fetal bovine serum (FBS) was obtained from Merck (Darmstadt, Germany). Cell culture inserts (pore diameter of 0.4 μm, PET, 1.12 cm^2^) were purchased from VWR (Leuven, Belgium). Caco-2 cell line, obtained from human colon carcinoma, was kindly provided by the Department of Biology of the University of Minho (Braga, Portugal). 

### 2.2. Nanoemulsion Preparation

Curcumin-loaded NEs were prepared through high-pressure homogenization according to other authors [37]. The lipid phase, constituted by LCT and 0.1% of curcumin, was homogenized with the aqueous phase (i.e., LEC, RHAM, or TWE at 2.5%) at room temperature and a volume ratio of 1:9. First, both solutions were pre-mixed using an Ultra-Turrax homogenizer (T18, Ika-Werke, Staufen, Germany) for 2 min and, thereafter, the resulting emulsion was passed through a high-pressure homogenizer (NanoDeBee, Bee International, South Easton, MA, USA) at 20,000 psi (137.9 MPa) for 20 cycles. The NEs were kept at 4 °C in the dark until particle characterization and in vitro digestion assays were performed.

### 2.3. In Vitro Digestion

In vitro digestion was performed using the static harmonized in vitro protocol that simulates the conditions of mouth, stomach, and small intestine [38]. Stock electrolyte solutions were prepared as follows. Simulated salivary fluid (SSF) was constituted by KCl 15.1 mmol L^−1^, KH_2_PO_4_ 3.7 mmol L^−1^, NaHCO_3_ 13.6 mmol L^−1^, MgCl_2_(H_2_O)_6_ 0.15 mmol L^−1^, (NH_4_)_2_CO_3_ 0.06 mmol L^−1^, and HCl 1.1 mmol L^−1^ in Milli-Q water. Simulated gastric fluid (SGF) was composed of KCl 6.9 mmol L^−1^, KH_2_PO_4_ 0.9 mmol L^−1^, NaHCO_3_ 25 mmol L^−1^, NaCl 47.2 mmol L^−1^, MgCl_2_(H_2_O)_6_ 0.12 mmol L^−1^, (NH_4_)_2_CO_3_ 0.5 mmol L^−1^, and HCl 15.6 mmol L^−1^ in Milli-Q water. Simulated intestinal fluid (SIF) was prepared with KCl 6.8 mmol L^−1^, KH_2_PO_4_ 0.8 mmol L^−1^, NaHCO_3_ 85 mmol L^−1^, NaCl 38.4 mmol L^−1^, MgCl_2_(H_2_O)_6_ 0.33 mmol L^−1^, CaCl_2_(H_2_O)_2_ 0.6 mmol L^−1^, and HCl 8.4 mmol L^−1^ in Milli-Q water. 

Briefly, the oral phase simulation consisted in the addition of SSF solution, CaCl_2_(H_2_O)_2_ 0.3 mol L^−1^ (in order to achieve 0.75 mmol L^−1^ at the final mixture), and Milli-Q water (in order to make up the final volume) to 5 mL of sample. The mixture was incubated for 2 min at 37 °C under agitation (120 rpm) at pH 7. Note that α-amylase was not used since the samples did not contain starch [39].

In the gastric phase, SGF, CaCl_2_(H_2_O)_2_ 0.3 mol L^−1^ (in order to achieve 0.075 mmol L^−1^ at the final mixture), and pepsin solution (with final activity of 2000 U mL^−1^ in the final mixture) were added. The pH was adjusted to 3.0 with HCl (1 mol L^−1^) and Milli-Q water was added to make up the final volume. The samples were incubated for 2 h at 37 °C under orbital agitation at 120 rpm.

The intestinal phase was simulated by adding SIF, CaCl_2_(H_2_O)_2_ 0.3 mol L^−1^ (in order to achieve 0.3 mmol L^−1^ at the final mixture), bile salts (in order to reach the concentration of 10 mmol L^−1^ in the final mixture), and pancreatin solution (with final activity of 100 U mL^−1^ in the final mixture). The pH was adjusted to 7.0 with NaOH (1 mol L^−1^) or HCl (1 mol L^−1^) and then Milli-Q water was added to make up the final volume. The samples were incubated for 2 h at 37 °C under orbital agitation at 120 rpm.

Samples were collected after each in vitro digestion phase (oral, gastric, and intestinal) and gastric phase reaction (pepsin activity) was stopped by putting the samples in an ice bath. At the end of digestion, the reaction was stopped by adding the enzyme inhibitor Pefabloc^®^ (1 mmol L^−1^) (10 µL for each 1 mL of sample). All the samples were tested at least in triplicate.

### 2.4. Particle Characterization

#### 2.4.1. Particle Size, Polydispersity Index, and ζ-Potential

The particles’ size and polydispersity index (PDI) were determined at each step of the digestion process using a dynamic light scattering instrument (Zetasizer Nano SZ, Malvern, Worcestershire, UK). The ζ-potential was also determined at each digestion stage using a particle electrophoresis instrument (Zetasizer Nano SZ, Malvern, Worcestershire, UK), which measures the particle velocity when subjected to an electric field. All the samples were diluted on a buffer solution (same pH of the samples) at a ratio of 1:100.

#### 2.4.2. Free Fatty Acid (FFA) Release

NE lipid phase digestibility was determined through FFA release method. The sample at the end of the gastric phase was mixed with all salts mentioned above (Section 2.3) in the intestinal phase and pH was adjusted to 7.0 with HCl (1 mol L^−1^). Then, pancreatin solution was added to the sample and pH was maintained at 7.0 by the addition of 0.05 mol L^−1^ NaOH solution using an auto-titration unit (pH-stat method) (Titrando 902, Metrohm, Switzerland) for 2 h in a heated jacketed reactor at 37 ˚C under agitation. At the end of the incubation at pH 7.0, NaOH was added to quickly reach pH 9.0, stopping the reaction and promoting FFA release. The FFA release was determined through the NaOH volume used to achieve pH 9.0 in order to guarantee full FFA ionization and titration [40,41]. Blank assays were performed (i.e., digestion conducted without pancreatin) to determine the NaOH volume needed to achieve pH 9.0. The amount of FFA released, *%FFA*, was calculated using Equation (1) [42]:(1)$$\%FFA=\left(\frac{\left(V_{NaOH\;sample}-V_{NaOH\;blank}\right)\times m_{NaOH}\times M_{lipid}}{w_{lipid}\times 2}\right)$$
where *V_NaOH sample_* and *V_NaOH blank_* are the volume of NaOH used to neutralize the FFA released in the sample and in blank assays, respectively; *m_NaOH_* is the molar concentration of NaOH titrant (0.05 mol L^−1^); *M_lipid_* is the corn oil molecular weight (it was considered 800 g mol^−1^ based on corn oil average fatty acid composition) [37], and *w_lipid_* is the total corn oil weight initially present.

#### 2.4.3. Curcumin Bioaccessibility and Stability

Curcumin’s bioaccessibility was assumed as the fraction of curcumin present inside the micelle phase, while stability was assumed as the fraction of curcumin present in the whole digesta at the end of the digestion. Curcumin bioaccessibility and stability were determined at the end of the digestion based on the methodology described by Liu et al. (2018) [43], with some modifications. The digesta (10 mL) was centrifuged (Allegra 64R, Beckman Coulter Inc., Brea, CA, USA) at 18,700× *g* at room temperature for 30 min and the supernatant was collected and assumed to correspond to the micelle phase. Digesta or micelle phase samples (5 mL) were mixed with 5 mL of chloroform using a vortex and centrifuged at 700× *g* at room temperature for 10 min. The bottom layer was collected and the top layer was subjected again to the extraction procedure. The second bottom layer was added to the first one and analyzed in a UV–VIS spectrophotometer (V-560, Jasco, Portland, OR, USA) at 420 nm. Curcumin concentration was determined through a calibration curve of absorbance versus curcumin concentration in chloroform.

The bioaccessibility (*B*), stability (*S*), and effective bioavailability (*BA*) were calculated using the following equations:(2)$$B=\frac{C_{Micelle}}{C_{Digesta}}\times 100$$
(3)$$S=\frac{C_{Digesta}}{C_{Initial}}\times 100$$
(4)BA=B×S
where *C_micelle_* and *C_Digesta_* are the curcumin concentrations measured at the end of the digestion in micellar phase and raw digesta, respectively. *C_initial_* is the curcumin concentration present in the NE at the beginning of digestion process. The effective bioavailability is an estimation of the curcumin absorption. However, this value must be analyzed with care, since there are other factors that influence curcumin bioavailability that are not considered, such as absorption and metabolism [44].

#### 2.4.4. Fluorescence Microscopy

The microstructure of the emulsions was observed using a fluorescence microscope (Olympus, BX51, Tokyo, Japan). The samples were stained with Nile Red (0.25 mg mL^−1^ in DMSO) at a ratio of 1:10 (dye: sample, *v*/*v*), which enabled the oil droplets to become visible. The images were captured from the initial to the final phase of the digestion process with a 100 × oil immersion objective lens.

### 2.5. Cytotoxicity Assays

Caco-2 cell line (ATCC) was grown in DMEM supplemented with 10% (*v*/*v*) FBS, 1% (*v*/*v*) non-essential amino acid solution, and 1% (*v*/*v*) PS. Cell culture was grown in a humidified 5% CO_2_ incubator at 37 °C. For the metabolic activity MTT assay, cells were cultured at 2 × 10^5^ cells/well in a 96-well microtiter plate. After 24 h of incubation at 37 °C in 5% CO_2_, the cell culture medium was replaced by 200 µL of fresh culture medium containing tested NE (5, 10, 15, and 25 µg mL^−1^) or free curcumin. After 4 h of incubation at 37 °C and 5% CO_2_, the medium was removed, and cells were washed with 200 μL of PBS. Then, 100 μL of MTT solution (0.5 mg mL^−1^ in PBS) was added to wells and incubated at 37 °C in 5% CO_2_ for 3 h. After this, 200 µL DMSO was added to each well to solubilize formazan crystals generated by live cells, followed by gentle stirring for 30 min on an orbital shake. Cell viability was assessed by spectrophotometry at 570 nm (reference wavelength 630 nm). Four replicates of each sample were analyzed. Cell viability is calculated from the following expression:(5)$$\mathrm{Cell}\;\mathrm{Viability}\;\left(\%\right)=\frac{Abs_{sample}}{Abs_{control}}\times 100$$
where *Abs_sample_* denotes the absorbance obtained from the wells containing treated cells and *Abs_control_* is the absorbance of untreated cells (cells in control medium).

### 2.6. Permeability Assays

Curcumin permeability studies were carried out based on the methodology proposed by Silva et al. (2019) [45], with some modifications. Caco-2 cells were cultured at 1 × 10^5^ cells/well on 12-well cell culture plate inserts for 18–21 d. The culture medium was changed three times a week. The cell monolayer’s development and integrity were controlled two to three times a week by the transepithelial electrical resistance (TEER) measurements using an epithelial Volt-Ohmmeter (Millipore Millicell ERS-2, Burlington, MA, USA). On the experiment day, the culture medium was removed; the cells were washed twice with 200 μL of HBSS and incubated for 15 min at 37 °C in 5% CO_2_ with 200 μL HBSS solution (transporter buffer). For the permeability experiment, 0.5 mL of NE (at 25 µg mL^−1^ of curcumin) diluted on HBSS was added to the apical side compartment. In order to maintain well conditions, 1.5 mL of HBSS with 1% of TWE was added to the basolateral side compartment [46]. At defined times (0, 15, 30, 45, 60, 120, 180, and 240 min), basolateral samples were collected. All samples were frozen until curcumin determination through HPLC-UV, as described in Section 2.7. 

The apparent permeability coefficients (*P_app_*) of curcumin were determined using the following Equation (6):(6)Papp=(dQdt)(C0×A)
where *dQ/dt* is the cumulative transport rate (μg min^−1^) across the monolayer established as the slope obtained by the linear regression of cumulative transported amount as a function of time, *C_0_* is the initial curcumin concentration (μg mL^−1^) in the apical compartment, and *A* is the surface area of the membrane (1.12 cm^2^).

### 2.7. HPLC Analysis of Curcumin

Curcumin concentration was determined based on the methodology described by Silva et al. (2019) [45]. The samples were diluted in acetonitrile at a volume ratio of 1:1 and centrifuged for 15 min at 14,000 rpm. After this, the supernatant was collected and injected into the HPLC system. The HPLC system was composed of a Varian Prostar 210 pump, a Varian Prostar 410 autosampler, and a Jasco FP-920 fluorescence detector (*λ_exc_* = 420 nm and *λ_em_* = 540 nm). A Varian 850-MIB data system interface and a Galaxie chromatography data system were used to manage the instrument and the chromatographic data. The HPLC separation was carried out on a C18 reverse-phase YMC-Pack ODS-AQ analytical column (250 × 4.6 mm I.D., 5 μm), fitted with a pre-column with the same stationary phase. The mobile phase consisted of a mixture of acetic acid (2% *v*/*v*) at pH 2.5 and acetonitrile at a volume ratio of 47:53, which was filtered and degassed with a 0.22-μm nylon membrane filter (GHP, Gelman). The compounds were eluted with a flow rate of 1.0 mL min^−1^ during a 15-min isocratic run at room temperature and the injection volume was 50 μL. The calibration curve was generated using standard solutions with concentrations between 0.1 and 10 μg mL^−1^ of curcumin in acetonitrile. The retention times of bisdemethoxycurcumin, demethoxycurcumin, and curcumin were 10, 11, and 12 min, respectively. Their quantification was performed through the comparison between the peak areas obtained and the calibration curve.

### 2.8. Statistical Analyses

All assays were performed at least in triplicate and presented as mean ± standard deviation (SD). Statistical analysis was carried out using OriginPro 2018 Statistic software (version b9.5.1.195; OriginLab Corporation, Northampton, MA, USA). Data were analyzed using one-way analysis of variance (ANOVA), and Tukey’s test was used to evaluate statistically significant differences between the mean values (*p* < 0.05).

## 3. Results and Discussion

### 3.1. Particle Characterization

All NEs were prepared using the same procedure, where the emulsifier type was changed and its concentration was maintained. NE characterization, namely particles’ size, PDI, and ζ-potential, was performed after their production, and the results are presented in Table 1. It is possible to observe that NE_RHAM presented the smallest particle size, followed by NE_TWE and NE_LEC (*p* < 0.05). These results can be explained by the interfacial properties of each emulsifier, which is considered an important factor in the ability to form and stabilize NE [47]. These results are in agreement with previous works. For instance, Liu et al. (2016) tested different emulsifiers (saponins, RHAM, and TWE) to develop emulsion-based fish oil delivery systems and reported that the particle size of emulsions produced with RHAM was lower than the size of emulsions produced with TWE. The authors also determined the interfacial tension and calculated the surface activity of each emulsifier, noting that RHAM had lower interfacial tension and higher surface activity than TWE, confirming that emulsions produced with RHAM need less energy to break up the droplets into small sizes [25]. Additionally, Arancibia et al. (2017) observed that LEC presented higher interfacial tension values than TWE and, therefore, the particle size of NEs produced with LEC was higher than those produced with TWE [47].

All NEs presented a narrow size distribution (PDI < 0.3), which is an indicator of good stability; see Table 1. NE_LEC presented a PDI value significantly higher than NE_TWE (*p* < 0.05), probably due to their interfacial tension. In terms of ζ-potential, all NEs showed values close to zero; see Table 1. NE_RHAM had a value that was more negative, followed by NE_LEC and NE_TWE. These results are related to the electrical charge of each emulsifier. RHAMs have anionic functional groups (i.e., carboxylic acid groups) that have a negative charge at neutral pH [25], TWE is a non-ionic emulsifier [47], and LEC, although being mainly composed of zwitterionic phosphatidylcholine (which is neutral at neutral pH), has other anionic phospholipids, such as phosphatidic acid, in its composition, which is negatively charged at neutral pH [48]. In general, NEs are considered stable when they present ζ-potential values around ± 30 mV. Despite the ζ-potential values observed being close to zero, NEs were shown to be stable at least during 6 weeks of storage (results not shown).

### 3.2. In Vitro Digestion

#### 3.2.1. Particle Characterization

The NE behavior in terms of particle size and ζ-potential was evaluated during in vitro gastrointestinal digestion; see Figure 2a,b, respectively. 

All NEs were shown to be stable at the oral phase; however, only NE_TWE maintained its stability at the gastric and intestinal phases since the value of particle size remained close to the one observed before the digestion process—see Figure 2a—although an increase in the PDI value was observed at the intestinal phase (results not shown). NE_RHAM presented the highest particle size increase at the gastric phase, from 155.5 ± 6.4 nm to 4041.1 ± 832.8 nm (*p* < 0.05)—see Figure 2a—and the highest increase in the PDI values (from 0.285 ± 0.040 to 0.969 ± 0.058). NE_LEC also presented a particle size increase from 184.1 ± 1.71 nm to 2063.3 ± 235.09 nm (*p* < 0.05) and an increase in the PDI from 0.246 ± 0.014 to 0.880 ± 0.097. However, at the intestinal phase, NE_LEC and NE_RHAM showed a decrease in particle size to values close to the ones observed at the initial stage, although with higher PDI values (0.539 ± 0.177 and 0.629 ± 0.110, respectively). 

Figure 3 shows that NE_RHAM displayed some coalescence at the gastric phase, whereas NE_LEC showed some flocculation and NE_TWE maintained a particle size close to the values observed at the initial phase. 

Regarding ζ-potential results, as shown in Figure 2b, all initial NE ζ-potential values increased from values close to zero to values between −15.7 mV and −84.3 mV at the oral phase, with NE_RHAM presenting the highest ζ-potential (*p* < 0.05). This could be due to the ionic species present in the salivary fluid, which can be adsorbed to the droplet surface, thus contributing to an increase in the surface charge. At the gastric phase, all NEs presented a decrease in the magnitude of ζ-potential to values close to zero, probably due to the decrease in pH and the increase in the ionic strength in the gastric fluids [49]. The exposure to pH 3 at 37 °C for 2 h, in the gastric phase, may have promoted the hydrolysis of LEC and RHAM lipid components, resulting in a negative charge (due to the production of FFA) that would attract the divalent cations present in the SGF (e.g., calcium and magnesium), leading to a roughly neutral ζ-potential that allows coalescence; see Figure 2a and Figure 3. In the case of NE_TWE, once they are stabilized by steric repulsion, they are not affected by the neutral ζ-potential (i.e., coalescence does not occur; see Figure 2a and Figure 3). At the intestinal phase, all NE had an increase in the magnitude of ζ-potential to values around −40 mV and −50 mV. This result can be due to pH increase or lipase and anionic component adsorption, particularly due to the presence of bile salts in SIF and lipid digestion products (e.g., FFA) [50].

Other authors also evaluated NEs’ size and ζ-potential produced using RHAM, LEC, and TWE as emulsifiers at different pH values and ionic strengths. Bai and McClements (2016) observed that NEs produced with RHAM were unstable at low pH (i.e., below pH 4) and 100 mM NaCl, showing an increase in particle size and a decrease in ζ-potential value. These authors showed that electrostatic repulsion is the main stabilization mechanism and that NEs are highly susceptible to salt-induced coalescence [26]. Gao et al. (2020) observed that NEs produced with LEC showed instability at low pH (i.e., pH 3) and 100 mM NaCl, because the particle size increased and ζ-potential value decreased. On the other hand, NEs produced with TWE maintained their particle size and ζ-potential values, being stable at all pH levels and ionic strength ranges tested. The instability shown by NEs produced with LEC can be due to the anionic groups present in the phospholipids, which lose their charge at low pH and in the presence of NaCl solution. NEs produced with TWE remained stable due to the steric repulsion [51].

Consequently, the type of emulsifier has a major impact on NE stability during the digestion process, where the stabilization mechanism is one of the main factors behind NE stability throughout the digestion process.

#### 3.2.2. FFA Release

At the intestinal digestion phase, all NEs exhibited fast FFA release during the initial minutes of the process, followed by a gradual FFA increase in the remaining time (data not shown). At the end, all lipids of NE_TWE were hydrolyzed into FFA (around 100%); see Figure 3. On the other hand, NE_RHAM showed a higher FFA percentage released (80.6 ± 2.42%) compared to NE_LEC (*p* < 0.05) (70.2 ± 3.40%), as can be seen in Figure 4. It was possible to observe that the emulsifier’s nature had a clear effect on lipid digestibility degree. Lipid digestion is a chemical interfacial reaction strongly dependent on the droplets’ particle size. Particles with smaller sizes have a higher specific surface area, increasing both oil phase availability and adsorbed lipase and/or bile salt interactions, consequently increasing the lipid digestion degree [51]. As discussed previously in Section 3.2.1, NE_TWE maintained its particle size over the digestion process at values close to the ones observed before digestion, whereas NE_LEC and NE_RHAM’s particle sizes increased during the gastric phase. Thus, NE_TWE’s specific surface area was higher than NE_LEC and NE_RHAM’s specific surface areas, which consequently led to a higher percentage of FFA released. Regarding the difference in the percentage of FFA released by NE_RHAM and NE_LEC, this may not be due to the droplets’ particle sizes, but to the properties of the emulsifiers. LEC is mainly composed of phospholipids, and it is known that phospholipids dispersed in an aqueous phase can affect lipase adsorption on the droplet’s surface as they can bind with pancreatic lipase, inhibiting lipase activity [52]. This may possibly explain the lowest percentage of FFA obtained for NE_LEC. Similar results have been obtained by other authors. For instance, Park, Mun, and Kim (2018) assessed the effect of different emulsifiers used in oil-in-water emulsions on lipid digestion and β-carotene bioaccessibility. The authors observed that emulsions produced with LEC showed the lowest FFA released when compared with emulsions containing Tween^®^ 20, whey protein isolate, soy protein isolate, and sodium caseinate [53].

#### 3.2.3. Curcumin Bioaccessibility, Stability, and Effective Bioavailability

Curcumin bioaccessibility and stability were determined at the end of the in vitro digestion process and curcumin’s effective bioavailability was estimated using bioaccessibility and stability values. NE_LEC showed the highest curcumin bioaccessibility, followed by NE_RHAM and NE_TWE (*p* < 0.05): 107.4 ± 12.57%, 41.8 ± 2.52%, and 33.2 ± 3.96%, respectively; see Figure 5. Regarding curcumin stability, NE_LEC presented similar values to NE_RHAM (27.6 ± 3.17% and 26.5 ± 1.67%, respectively), which were not statistically different (*p* > 0.05), and NE_TWE showed by far the lowest value (8.2 ± 0.66%); see Figure 5. Thus, NE_LEC had the highest curcumin effective bioavailability value, followed by NE_RHAM and NE_TWE: 29.4 ± 2.96%, 11.0 ± 0.36%, and 2.72 ± 0.30%, respectively (*p* < 0.05, Figure 5). Although NE_LEC presented the lowest FFA release percentage, it showed the highest curcumin bioaccessibility, which may be due to the emulsifier’s nature. It is known that the mixed micelles produced in the intestinal phase are mainly composed of bile salts, phospholipids from bile, and FFA. LEC is composed mainly of phospholipids, which may have facilitated mixed micelle formation and may have improved their solubilization capacity [35]. In contrast, NE_TWE presented the highest percentage of FFA released but the lowest curcumin bioaccessibility and stability values. These results are in agreement with those of Zou et al. (2015), who evaluated the effect of excipient emulsions with different emulsifiers (caseinate, whey protein isolate, and TWE) on curcumin’s solubility, stability, and bioaccessibility. Those authors observed that the excipient emulsions with TWE presented high FFA release; however, these excipient emulsions presented the lowest curcumin amount in raw digesta (i.e., lowest curcumin stability) and lowest curcumin quantity in the mixed micelle phase (i.e., lowest curcumin bioaccessibility) [54].

Therefore, it is clear that FFA production is not the most important factor for curcumin bioaccessibility and that the emulsifier type also has a strong impact on this property. This is probably due to emulsifier interactions with curcumin, with lipid digestion products, or with the enzymes present in the intestinal phase. Additionally, the components present in the emulsifier seem to have a marked effect on mixed micelle formation. 

### 3.3. Cellular Studies

The effect of the curcumin-loaded NEs produced with different emulsifiers on cell viability was determined and is reported below. The curcumin concentration range used in cellular studies was selected based on the curcumin concentration detected at the end of the in vitro digestion process. Cell viability tests were also performed with free curcumin (at the same concentration range used for NEs) and it was observed that free curcumin had no cytotoxic effects at that concentration range (data not shown). Moreover, it was possible to observe that NE_RHAM were highly cytotoxic to cells, whereas NE_LEC and NE_TWE showed no cytotoxicity at all concentrations tested; see Figure 6. Perinelli et al. (2017) studied the effect of RHAM on Caco-2 and Calu-3 cell viability and they observed that RHAM presented a cytotoxic effect for concentrations above 150 μg mL^−1^ [29]. In our work, NE presents a RHAM concentration of 1.125 mg mL^−1^ at the lowest curcumin concentration tested (i.e., 5 μg mL^−1^), which is higher than the maximum concentration presented by Perinelli et al. (2017) [29]. These results show that the emulsifier type and concentration used have a strong effect on cell viability. Moreover, it was shown that it is crucial to determine the potential cytotoxic effect of nanostructures even if they are only composed of bio-based, GRAS ingredients. As NE_RHAM presented cytotoxicity at all concentrations tested, it was not included in the cellular permeability assays. For NE_LEC and NE_TWE, the *P_app_* was determined at the highest curcumin concentration tested in the cell viability assay (i.e., 25 μg mL^−1^). NE_LEC and NE_TWE presented *P_app_* values of 6.59 × 10^−5^ ± 5.18 × 10^−6^ cm s^−1^ and 5.49 × 10^−5^ ± 2.80 × 10^−5^ cm s^−1^, respectively. No statistically significant differences (*p* > 0.05) in *P_app_* values between NE_LEC and NE_TWE were observed, meaning that the emulsifier type had no significant effects on the intestinal permeability of curcumin. Sessa et al. (2014) carried out permeability tests on Caco-2 cells for NEs with resveratrol produced with different emulsifiers and different particle sizes. Those authors observed that the particle size was the main factor influencing resveratrol permeability across Caco-2 monolayers. Additionally, the emulsifier used in the NE formulation had an effect on resveratrol permeability, where LEC promoted its permeability. These authors justify the results with LEC’s (i.e., composed by a phospholipid mixture) similarity with the bilayer structure of the cellular membrane, which can improve the compound transport through the epithelial cell membrane [55].

## 4. Conclusions

The influence of emulsifier type has been evaluated regarding NE behavior, curcumin bioaccessibility, and stability after in vitro digestion, as well as NEs’ effect on cell viability and curcumin permeability across Caco-2 cells. Although NE_TWE particle size was maintained during in vitro digestion and NE_TWE showed the highest FFA percentage released, this NE presented the lowest curcumin bioaccessibility and stability. On the other hand, NE_LEC presented the highest curcumin bioaccessibility, despite their instability at the gastric phase and the lower FFA percentage released. NE_RHAM presented a curcumin stability value similar to that of NE_LEC and a curcumin bioaccessibility value higher than that of NE_TWE. However, NE_RHAM were cytotoxic to cells at all concentrations tested, which was an unexpected result since this NE is composed only of bio-based and GRAS compounds. Our results showed that the emulsifier type has a strong impact on curcumin bioaccessibility and stability during digestion and on cell viability. Overall, NE_LEC was the NE formulation that yielded the most interesting results regarding both safety and functionally (i.e., curcumin bioaccessibility and stability). Furthermore, it is shown that it is crucial to evaluate nanostructures’ cytotoxicity, even if they are composed of GRAS and bio-based ingredients. This work gives new and important insights into ingredient selection for the development of safe and effective curcumin-loaded NEs for food applications.

## Figures and Tables

**Figure 1 nanomaterials-11-00815-f001:**
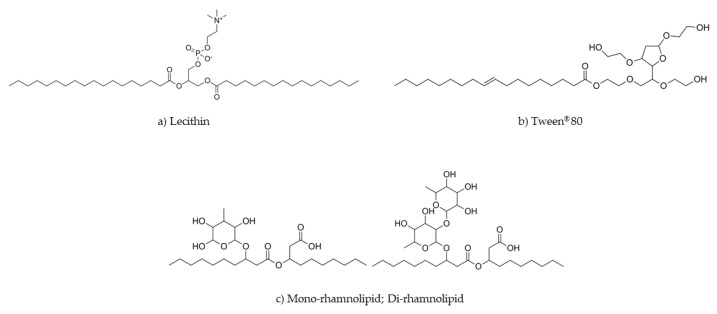
Chemical structure of the different surfactants used: (**a**) lecithin (LEC); (**b**) Tween^®^ 80 (TWE), and (**c**) two of most common rhamnolipids available (RHAMs).

**Figure 2 nanomaterials-11-00815-f002:**
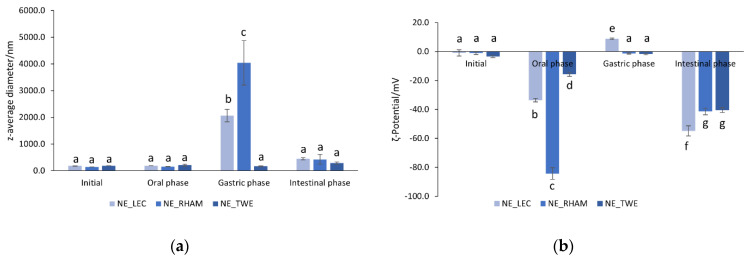
Particle size (**a**) and ζ-potential (**b**) of nanoemulsions (NEs) with different emulsifiers (lecithin (LEC), rhamnolipids (RHAMs), and Tween^®^ 80 (TWE)) in the different in vitro gastrointestinal digestion phases. Error bars represent the standard deviation of *n* = 3 replicates. ^a–g^ Different lower-case letters indicate a statistically significant difference between NEs in the different GI) phases (*p* < 0.05).

**Figure 3 nanomaterials-11-00815-f003:**
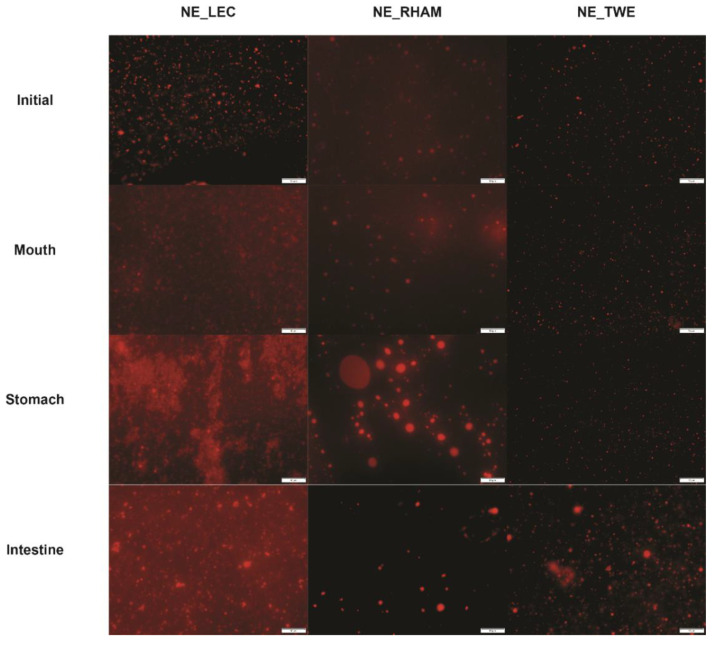
Fluorescence microscopy images of nanoemulsions (NEs) with different emulsifiers (lecithin (LEC), rhamnolipids (RHAMs), and Tween^®^ 80 (TWE)) after each in vitro gastrointestinal digestion phase. The scale bar for all images is 10 μm.

**Figure 4 nanomaterials-11-00815-f004:**
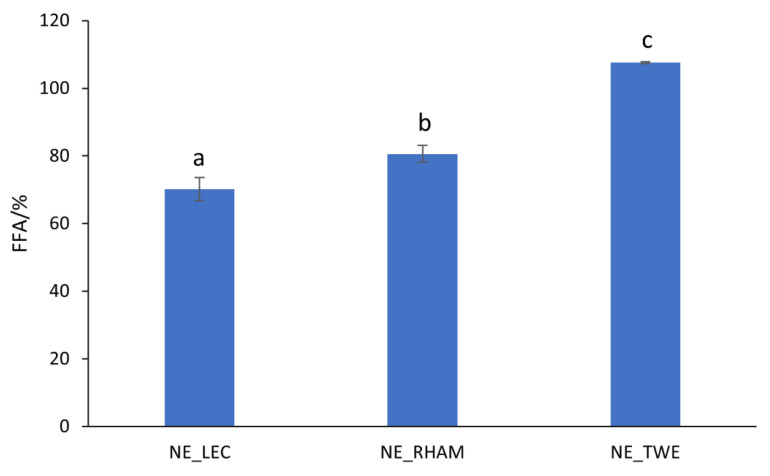
Percentage of free fatty acids (FFA) released after in vitro digestion process of curcumin-loaded nanoemulsions (NEs) with different emulsifiers (lecithin (LEC), rhamnolipids (RHAMs), and Tween^®^ 80 (TWE)). Error bars represent the standard deviation of *n* = 3 replications. ^a–c^ Different letters indicate a statistically significant difference between NEs (*p* < 0.05).

**Figure 5 nanomaterials-11-00815-f005:**
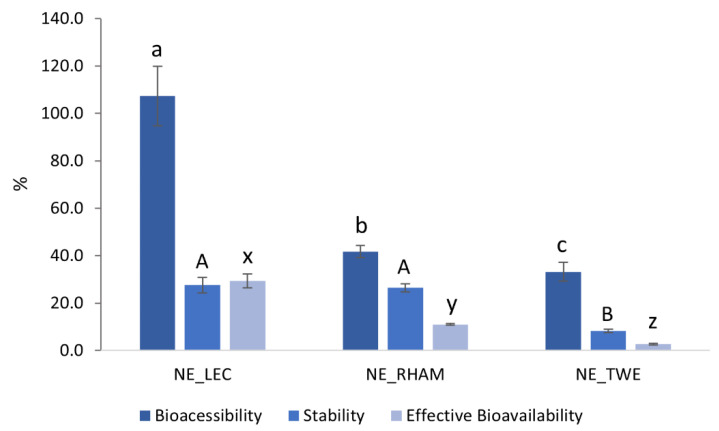
Bioaccessibility, stability, and effective bioavailability of curcumin loaded in nanoemulsions (NEs) with different emulsifiers (lectithin (LEC), rhamnolipids (RHAMs), and Tween^®^80 (TWE)) after in vitro digestion process. Error bars represent the standard deviation of *n* = 6 replications. Different letters indicate statistically significant differences between NEs in ^a–c^ bioaccessibility, ^A,B^ stability, and ^x–z^ effective bioavailability (*p* < 0.05).

**Figure 6 nanomaterials-11-00815-f006:**
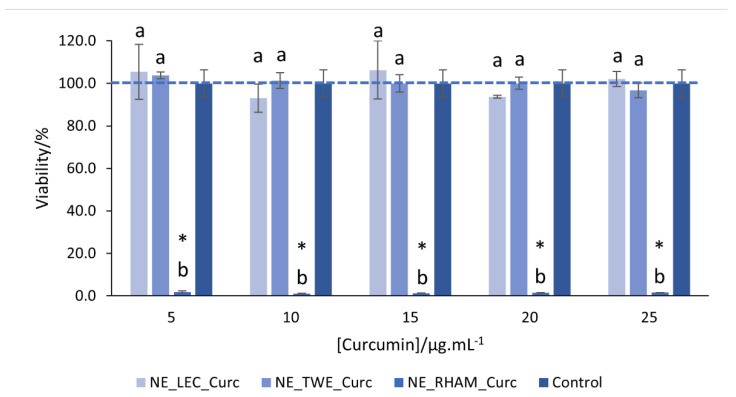
Effect of curcumin-loaded nanoemulsions (NEs) produced with different emulsifiers (lecithin (LEC), rhamnolipids (RHAMs), and Tween^®^ 80 (TWE)) on viability of Caco-2 cells. Error bars represent the standard deviation of *n* = 3 replications. ^a–b^ Different letters indicate a statistically significant difference between NE/concentration interactions (*p* < 0.05). *Asterisks indicate statistically significant differences relative to the control group (*p* < 0.05).

**Table 1 nanomaterials-11-00815-t001:** Particle size, polydispersity index (PDI), and ζ-potential of curcumin-loaded nanoemulsions (NEs) produced with different emulsifiers (rhamnolipids (RHAM), lecithin (LEC), and tween 80 (TWE)).

Nanostructure	z-Average Diameter (nm)	PDI	ζ-Potential (mV)
NE_RHAM	144.1 ± 3.7 ^a^	0.192 ± 0.031 ^a,b^	−2.45 ± 2.75 ^a^
NE_LEC	176.1 ± 6.1 ^b^	0.222 ± 0.014 ^a^	−1.49 ± 2.59 ^a^
NE_TWE	167.4 ± 5.3 ^c^	0.191 ± 0.024 ^b^	1.95 ± 2.19 ^b^

^a–c^ Mean values with different superscript letters within the same column are significantly different from each other (*p* < 0.05).

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
