# Peer review of "Nanoemulsions for Enhancement of Curcumin Bioavailability and Their Safety Evaluation: Effect of Emulsifier Type"

_nanomaterials, 2021, doi:10.3390/nano11030815_

Round 1

Reviewer 1 Report

Regarding: “Nanoemulsions for enhancement of curcumin bioavailability and their safety evaluation: effect of emulsifier type” by Goncalves et al, submitted for publication in nanomaterials.

This submission describes the characterization of nanoemulsions (NE) formulated from either lecithin (LEC), rhamnolipids (RHAM) or Tween80 (TWE) lipids, with the goal of evaluating their safety and efficacy as delivery vehicles for the health supplement curcumin (and by extension other therapeutics of low water solubility.)

The various NE were characterized first in terms of their respective particle size and zeta-potential upon being exposed to simulated salivary fluid (SSF), simulated gastric fluid (SGF) and simulated intestinal fluid (SIF).  Free fatty acid release after exposure to SGF was also measured.  Ultimately, curcumin bioaccessibility was assessed by measuring curcumin remaining in the micellar phase of the NE. Finally, the cytotoxicity of the various NE and their ability to deliver curcumin into cells were evaluated in cell cultures.

Table 1 presents results concerning particle size and zeta potential results for the three NE’s before exposure to the various simulated digestive fluids.  All three NE’s exhibited narrow particle size distributions and relatively small zeta-potentials.  Concerning the negative zeta-potential found for LEC NE, the authors state (line 294) that “LEC is an amphiphilic phospholipid which also presents negative charge at neutral pH…..”  This is confusing, since the major component of LEC is zwitterionic phosphatidylcholine, which is neutral at neutral pH.  The authors should reference the point that, in fact, the LEC likely contains other phospholipids, including in particular phosphatidic acid, which is negatively charged at neutral pH.

Figure 1 shows particle size and zeta-potential data for all three NE’s after exposure to one of the three simulated digestion fluids.  Only the SGF had any significant effect on particle size, (Fig 1A) which increased radically but only for the LEC NE and RHAM NE, and not the TWE NE.  The principal difference between SGF exposure (pH 3 for 2 hours at 37 C) and SIF (pH 7 for 2 hours at 37 C) or SSF (pH ? for 2 minutes at 37 C) is obviously pH.  I would argue that exposure to pH 3 at 37 C for 2 hours is highly likely to produce significant hydrolysis of the fatty acyl esters of the LEC and RHAM lipids (but less so for TWE). The resulting negative charge on free fatty acids would attract the divalent cations in the SGF (calcium and magnesium) leading to a roughly neutral zeta-potential that permits particle coalescence. The TWE NE being stabilized by steric repulsion are not affected.  Since that authors evaluate fatty acid release from the various NE in a later section, it would be important to establish how much free fatty acid was present immediately after exposure to SGF.  I believe the authors may have some data relevant to that question, and if so they should reveal it.

As to the zeta-potential effects shown in Fig. 1b, the large negative zeta-potential of all three NE’s upon exposure to the SIF must surely be attributed specifically to the presence of bile salts in that latter fluid (versus their absence in the others.) Bile salts will specifically associate with the micellar phase.  The authors should state this up front.  As an aside, I could find no mention of which specific bile salts were employed nor where the authors purchased the bile salts used here.  (For that matter it is never specified what the final pH was for the SSF. Also, it would be useful to know the final osmolarity of all three simulated digestion fluids, perhaps in a table.)

The results concerning free fatty acid release, curcumin bioavailability and bioaccessibililty appear reasonable.  The most surprising result for this reviewer was the cytotoxicity of the RHAM NE (Fig. 5).  However, as the authors note, this result conforms with an earlier report from Perinelli et al (2017). 

Overall, this submission represents a useful addition to the understanding of NE as delivery vehicles.  The text is admirably free of errors, while the figures and tables are clear and appropriate.  Provided the authors address the various points I raise above, I am comfortable recommending publication in nanomaterials.

Reviewer 2 Report

This study deals with the curcumin bioavailability mediated by nanoemulsions tested by means of an in vitro digestion system and a cytotoxicity study. Specifically, the manuscript regards the interesting topic related to the fabrication of stable matrices for the huge area of food applications. The authors in continuity with their important previous contribution in the field provided a further evaluation of the digestion/absorption behaviour of curcumin nanoemulsions produced using various emulsifiers.

In this direction the level of novelty is mainly related with the utilization of rhamnolipids (less studied as stated from the authors), other information of these systems in the presence of other surfactants were always present in literature.

The manuscript is well written and interesting.

However, there are some weakness in the whole manuscript that should be necessarily clarified, some details should be better addressed and specified for the sake of clarity of the readers:

- Considering the readership of the journal (Nanomaterials, not all readers are experts in the field) further details should be provided regarding the “material” nanoemulsions and surfactants...may in terms of schematic representation and chemical structures.

- A deeper analysis concerning the introduction section could/should be provided, if as the authors claim this manuscript should represent a contribution to the state-of-the-art’s progress. As regard, several recent contributions referring to the same field and operating with the same approach, (curcumin bioavailability, curcumin delivery systems in the presence of Tween 80, LEC based systems, in vivo digestion, role of charge, pH, role of fluid constituents and so on…) were completely neglected. These information mays could help in improving the message in the discussion and the conclusions and could justify the level of novelty of the present paper. This piece of information could be better addressed in the introduction section.

- As for the z-potential study, by analysing the digestion process the authors refers only to the lipase activity. What about other enzymes activity present in the different bulk of the fluids?

- Fig 1 A (DLS): what is the pdI value of the reported samples?

- Fig 2: where are the comments of figure? The readers in my opinion are not able to understand the significance of these information.

- The authors must provide the relevance of the study and the ways in which the new results have advanced the field in comparison with the previous papers (and already reported results). In this respect the authors should provide specifications: if the novelty of this paper mainly regards the role of rhamnolipids this should be clearly reported starting from the title of the manuscript and distinguishing new results with previous consolidated one. This piece of information should be appropriately addressed in the whole manuscript and particularly in discussion section.

As a whole, apart from all, in my opinion if the conceptual novelty to the readers of the specific message will be accurately provided in each section of the manuscript and aims and scope of the journal will be satisfied, the manuscript can be considered for publication.

Round 2

Reviewer 2 Report

The manuscript has been improved and now could warrants publication in Nanomaterials.

Best regards.